# Empowerment-Based Physical Activity Intervention for People with Advanced Dry Age-Related Macular Degeneration: Mixed-Methods Protocol

**DOI:** 10.3390/ijerph20010643

**Published:** 2022-12-30

**Authors:** Eva-Carin Lindgren, Jeanette Källstrand, Åsa Alftberg, Pia Johansson, Lars Kristén, Linn Håman, Andreas Ivarsson, Ing-Marie Carlsson

**Affiliations:** 1School of Health and Welfare, Halmstad University, SE-30118 Halmstad, Sweden; 2Department of Social Work, Faculty of Health and Society, Malmö University, SE-20506 Malmö, Sweden

**Keywords:** adapted physical activity, age-related macular degeneration, empowerment, intervention, mixed methods

## Abstract

Age-related macular degeneration (AMD) is the most common cause of incurable visual impairment and impacts daily life. These impacts include loss of social activities, decreased functional independence, and reduced physical activity. This protocol aims to describe a prospective, mixed-methodology for studying a population with AMD before, during, and after an empowerment-based physical activity intervention (EPI). A study framework was also developed for EPI. The intervention will include 20 older individuals (age 65+ years) with AMD recruited in Sweden. The intervention period is six months and comprises adapted physical activity and social activities in a group twice a week and individual health coaching on three occasions. The quantitative pre-test and three follow-ups include physical functional tests, an accelerometer that monitors physical activity continuously for one week, and questionnaires. Individual and focus-group interviews and ethnographic observations will explore the experience of living with AMD and what it means to participate in the EPI for individuals with AMD. The chosen methodology offers a structured way for researchers to explore the experiences and factors that may provide insights into the potential of creative supervised, adapted physical activity in groups, health coaching, and socialising that are significant to enable well-being among older individuals with AMD.

## 1. Introduction

Age-Related Macular Degeneration (AMD) affects an individual’s central vision and is considered the third most common cause of impaired vision and legal blindness among people 60 years and older [1,2]. In 2020, 200 million people were estimated to have AMD, and the number is expected to increase to as many as 300 million by 2040 [2] and even further as the proportion of older people increases [3]. According to the Framingham Eye Study, the prevalence of AMD was 1.6% in those aged 52–64 years, 11% for 65–74 years, and 27.9% for 75–85 years [4]. In Scandinavia, the prevalence of early AMD was 3.5% for those aged 55–59 years and 17.6% for those aged 85 years and older, and because of advanced AMD, 0.1% and 9.8%, respectively [5]. The rates of advanced AMD are estimated to increase by 75% between 2012 and 2040, corresponding to 328,000 people, of which 97,000 people live in Sweden [5].

AMD occurs in two forms, wet (neovascular) and dry (atrophic), and may affect one or both eyes but causes no pain [1,6]. Independently of the form, the central vision is affected, mainly including detail, contrast sensitivity, and colour vision, but the peripheral navigational vision remains. Peripheral visual functions in people with mild visual deficits may improve through psychophysical training such as Perceptual Learning (PL), a method used after different kinds of sensory damage [7]. Nevertheless, the effect of PL in advanced AMD seems to be less effective. Still, improvements in elementary visual functions do occur such as some types of visual acuity, orientation discrimination, contrast sensitivity, and visual crowding is reduced [8]. Yet, central vision is still affected.

In the early stages of AMD, an individual may not notice the gradual loss of vision until it is evident that central vision is affected [9]. When AMD becomes severe, it often causes deterioration in the quality of life because of limitations when performing everyday life activities [10,11,12,13]. It is known that visual impairment often has a serious impact on an individual’s life since the visual ability is of great importance in regard to independence, active lifestyle, and above all, quality of life [10,11,12,13].

Social interactions are essential to daily life, and face recognition is fundamental because of both its communicative and reflexive components [1]. Depending on the severity of the disease, people with AMD have difficulties with different aspects of face recognition, such as the inability to interpret facial expressions [14,15]. Feelings of anxiety of being embarrassed occur in situations such as not recognising friends when meeting them [1,16]. Therefore, visual impairment is a limitation regarding participation in social activities, visiting various events, and meeting friends, which can lead to social isolation [10,11,12,13].

The impact of AMD on distance assessment and perceiving contrasts makes it challenging to detect hazards in the environment and can lead to problems with balance ([17,18]). These obstacles may result in impaired physical activity with subsequently decreased muscle strength and further problems with balance, and an increased risk of unintentional falls and fall injuries. In the long run, visual impairment can lead to mental illnesses such as depression [13,19,20]. However, better functioning can be achieved through social support, contact with low-vision clinics and aid to avoid fatalistic attitudes regarding undesirable effects of ageing, recent vision loss, fear of further vision loss, and social isolation [19,21].

Research shows that physical activity reduces co-morbidity and improves the quality of life [22], as well as mobility, movement patterns, and balance, thus reducing the risk of falls [23,24,25,26]. Overall, research on older people shows that regular physical activity positively affects their health.

Visual acuity, visual field, contrast sensitivity, and perceived visual ability are essential predictors when safely navigating one’s surroundings [27]. Older individuals with a visual impairment such as AMD experience various barriers regarding physical activity [28], such as fear of falling, which could lead to decreased physical activity and a less active lifestyle compared with other people of the same age who have a normal vision [29,30,31,32]. However, a systematic review shows that physical activity is associated with lower odds of early and especially late AMD, with odds ratios of 0.92 and 0.59, respectively, in white populations [33].

To create conditions for equal health, more knowledge is needed about the impact of dry AMD on daily life. Therefore, it is essential to investigate how people with AMD perceive everyday life and how physical activity coupled with social interaction can be designed to achieve a sustainable community so that health and well-being can be maintained despite vision loss. This study protocol aims to describe a prospective, mixed-methodology for studying older people with AMD before, during, and after an empowerment-based physical activity intervention. The research project will contain multiple studies, including qualitative data on the participants’ experiences and quantitative data on physical activity levels and self-reported quality of life.

### 1.1. Empowerment-Based Physical Activity Intervention

#### 1.1.1. The Approach

This research project will focus on a person’s health, not the disease (i.e., AMD) [34]. The intervention used is the Empowerment-Based Physical Activity Intervention (EPI), which will embrace empowerment as a process and adopts the reflective equilibrium community empowerment approach. This method combines top-down and bottom-up empowerment approaches [35]. The top-down approach means that the researchers identify the issue of concern and create an intervention to solve it. The bottom-up approach means that the participants gain power in the decision-making process. Specifically, the process aims to increase participants’ awareness of their own needs and wishes, which can play an active role in their everyday lives.

Participation in EPA should also strengthen the participants’ perceived ability to participate in physical activities. Participation in EPA is meant to support individuals’ opportunities to take control of their lifestyle [36] and thus be the ones who decide on what they want to train for increased sustainable physical activity and in what way to promote their health, well-being, and vision-related quality of life. Three health coaching sessions are planned to achieve this: at the beginning of the intervention, after four months, and after the end of the intervention (after six months). Health coaching is based on open-ended questions, exploring options, reframing, expressions of empathy, summaries, and goal setting [37,38].

The practitioners/researchers of the activities will communicate and interact with the participants twice a week and one hour after workouts. Then, they can listen to the participants’ needs and wishes, a natural process of exchange, dialogue, and negotiation between the participants and practitioners [35]. The purpose of health coaching is to view the participants as creative, resourceful, and capable of finding unsuspected solutions to their problem formulations and is in line with the empowerment approach as people are considered to possess the necessary means to change and develop in the desired direction [39].

Through collaboration with different organisations, participants can have a significant opportunity to continue participating in adapted physical activities after the end of the project. Thus, there is sustainability in what the project initiates. The collaboration will also involve sharing knowledge and understanding of AMD and the possible support needed for a visual impairment during physical activity.

#### 1.1.2. Activities

The activities in the EPI consist of adapted physical activities based on the participant’s abilities, social gatherings, and health coaching. Participants will be invited to participate in physical activities in balance training, which includes strength training, movement training, and flexibility training in groups for about one hour twice a week for six months, organised by the research project at Halmstad University’s health lab. This exercise aims for better safety and reduces the risk and fear of falls.

Additionally, offered are complementary activities led by the sports movement and adapted sports activities, such as table tennis for people with visual impairments. These will emphasise participation, new skills, and enjoyment rather than focusing on performance as in typical sports clubs. After the group training, social gatherings are offered for an hour, at which coffee, sandwich, and fruit are offered. Two undergraduate students and two researchers will carry out the physical activities, some sports activities, and the social gathering. The students will also implement health coaching. Researchers, coaches from sports associations, and activity leaders from Halmstad municipality will be invited to some social gatherings. Participants are offered at a later stage of the intervention; participants are offered a chance to participate in Halmstad municipality’s physical activity balance training via drop-in gyms arranged at various places in the city, where activity leaders are available and supported. The researchers will give a short lecture on AMD and provide an accelerometer, and the trainers and activity leaders will talk about their activities and invite the participants. Participants are offered at a later stage of the intervention to participate in Halmstad municipality’s physical activity balance training via drop-in gyms arranged at various places in the city, where activity leaders are available and supported.

## 2. Materials and Methods

### 2.1. Setting and Reference Group

EFA is based on an Interreg project, “Can you see the future?” and is a cross-border collaboration between Sweden and Denmark. However, the study protocol focuses only on the part of the project that will be carried out in a community (about 100,000 inhabitants) in the southwest of Sweden (i.e., EFA).

The project has established a reference group, which was invited to events where the project was presented with the possibility of engaging in knowledge exchange to ensure relevance. The focus was on describing the project idea, obtaining feedback on the implementation and evaluation of this intervention, and identifying potential pitfalls and challenges and how to overcome them. The reference group consisted of visually impaired people and professionals such as low-vision therapists, ophthalmic nurses, Halmstad Municipality activity centres, Halland’s Sports Movement, and Parasport Association employees. A consultation with a physiotherapist was also established to discuss which indicators to choose for physical tests.

### 2.2. Participants

The project will include eligible individuals with age ≥ 65 years who live in their own homes and have the cognitive ability to participate in the studies. The individuals must have a diagnosis of dry AMD in one or both eyes, visual acuity of no more than 0.3 in the best eye (0.8–1.0 corresponds to normal vision) and contact with a low vision clinic. The participants need to walk independently with or without walking aids. Furthermore, the person needs to understand and communicate freely in Swedish.

Candidates will be excluded if they have wet AMD, are undergoing treatment, or lack independent mobility. People with a disease that can pose a severe risk during physical activity will be excluded.

### 2.3. Recruitment Procedures

Responsible low-vision therapists and other staff within the low-vision clinic in Halland County are informed orally and in writing by researchers about the study’s purpose and structure. The team at the low-vision clinic identifies individuals based on inclusion criteria. Recruitment and inquiries about participation occur during a regular reception visit at the low-vision clinic or when staff contacts an individual by telephone and asks about participation.

### 2.4. Ethical Considerations

The Swedish Ethical Review Authority approved an ethics application for this study (EPN 2021/02877). The participants will receive oral and written information about the study. The low vision centre will send out information letters and invitations to participate. After about a week, responsible researchers contact the individuals to enable reflection time. Information will be given that participation is voluntary and can be interrupted without explanation and without affecting the participants’ continued care and treatment. Participants also receive information about personal data management according to GDPR, how the participants’ confidentiality is ensured, data management, and assurance that no one outside the research group will have access to data.

If there is interest in participating, an informed consent form is signed. All data will be coded so that only relevant researchers can access it. Data will be locked in a safe at the university, away from the code key, which will be closed in another safe. Research material is saved for ten years after the research principal completes the project. Only researchers who belong to the project will have access to data.

All information that can reveal the identity of the researchers will be pseudonymised. Participating researchers have solid experience with how this can be achieved. No increased risks for the participants have been identified as the project does not impair the participants but provides increased opportunities for increased physical activity and social community. Thus, participation is not considered to involve risks to the participant’s health, safety, or personal integrity.

### 2.5. Data Collection and Data Sources

Data will be collected before, during, and after the intervention with varying data sources (Figure 1):

Individual qualitative interviews with open-ended questions will be performed before the participants join the intervention, and ethnographic observations will be performed during the intervention. After the intervention ends, focus-group interviews will be performed with the participants (after six months).

Quantitative baseline data will be collected from recruited participants, including demographic data (age, sex, marital status, and housing). A test protocol for the quantitative data is used for the data collection at baseline, at four months, at the end of intervention at six months, and 12 months after the end of the intervention. The protocol comprises physical functional tests for the assessment of strength with the Sit and Stand Test (i.e., 30STS), Time Up and Go (TUG), and an accelerometer (3-axis accelerometer) worn for one week. Questionnaires for the self-reported assessment of visual functioning/vision-related quality of life (i.e., NEI VFQ-25), health-related quality of life (i.e., EQ-5D-5L), physical activity (i.e., PASE), and self-efficacy for exercise (i.e., EXSE) are included in the test protocol.

#### 2.5.1. Interviews

Individual and focus-group interviews will be conducted using a constructive grounded theory [40]. A constructivist grounded theory views reality as fundamentally social and processual [41]. According to the grounded theory method, data collection and analysis will co-occur [40]. The individual interviews will be performed before the intervention, and the focus-group interviews will be performed at the end of the intervention. A constructivist grounded theory views reality as fundamentally social and processual [41]. An individual interview can provide increased knowledge and understanding to promote physical activity, health, and the well-being of people with AMD. This activity starts an empowerment process involving listening to their opinions and experiences. The focus group interviews can contribute knowledge about what it means for people with AMD to participate in the EPI, increase their physical activity for their health and well-being, and how physical activities can be designed and implemented in the best way for increased sustainability for an individual.

#### 2.5.2. Observations

Participant observations will be performed regularly during the intervention sessions. The aim is to explore the social interaction within the group and the meaning-making of physical and social activities. This will provide in-depth knowledge of what enables sustainable participation in physical activity for people with AMD.

#### 2.5.3. Sit and Stand Test (30STS)

The 30STS [42] measures lower-extremity muscle strength. The participant will be asked to repeat as many sit-to-stand actions as possible in 30 s, and the score is the total number of stands completed within that time. Following oral instructions, a test rehearsal is performed before timekeeping starts. The only material needed is a chair with a standardized height (45 cm) and a stopwatch. The participants are encouraged to complete as many full stands as possible within 30 s with the arms crossed in front of the chest and the feet parallel. The test has good validity and reliability in measuring bone strength in the elderly [42].

#### 2.5.4. Timed up and Go

Dynamic balance ability will be assessed with the Timed Up and Go (TUG) test [43]. An examination of essential functional mobility has been shown to predict the ability of older people to walk outdoors independently. After instructions, the participants will start seated in a chair and then be asked to stand up and walk three meters, turn around, walk back to the chair, and sit down. The outcome measure is the time needed to complete the test. The TUG test has shown good reliability and validity for older people [43,44].

#### 2.5.5. Actigraphy Watch

The participants will wear an actigraphy watch on the wrist for seven days, according to previous recommendations [45]. Data will be processed in 60-s epochs, which be each categorised into one of four categories based on the type of activity that the participants engage in. The four categories of behaviours are (a) sedentary, (b) light-intensity physical activity, (c) moderate-intensity physical activity, and (d) vigorous physical activity.

#### 2.5.6. Visual Functioning Questionnaire

The National Eye Institute 25-item Visual Functioning Questionnaire (NEI VFQ-25) is a validated questionnaire that was developed to examine different dimensions of perceived vision-related quality of life associated with five chronic eye diseases such as age-related cataracts, AMD, diabetic retinopathy and primary open-angle glaucoma, and low vision from any cause [46,47]. The questionnaire has shown good psychometric properties in measuring visual function-related outcomes in individuals with AMD [45,48]. This project will use a translated Swedish version validated in the Early Manifest Glaucoma Trial [49] and used in other Swedish studies with a population older than 70 years [49,50].

The NEI VFQ-25 contains various statements about problems related to a participant’s vision or feelings regarding different conditions where vision is essential [47]. If participants usually wear glasses or contact lenses, they should respond as if they wear those. The questions deal with both health and vision from a general perspective, the presence of pain in or around the eyes, activities that require vision both near and far, social activities, mental health, participation in various activities, addiction, driving, colour vision, and peripheral vision [46]. The general health question is used in many studies as it is a reliable predictor of future health and mortality [47]. The NEI VFQ-25 score is calculated by producing an average of the vision-related subscales where the general health question is excluded [47].

#### 2.5.7. Health-Related Quality of Life

The EQ-5D-5L instrument is used to measure changes in health-related quality of life. The instrument is often used to describe the quality of life of patient groups and evaluate the effects of interventions in healthcare in Sweden and internationally [51]. The EQ-5D-5L is a generic instrument consisting of five questions with five answer options for each question. Based on the participant’s answers, a preference-based value between 0 and 1 called a quality-of-life weight, can be calculated. The Swedish value set [52] will be used. The quality-of-life weight can then be used to calculate quality-adjusted life years (QALYs) to measure health effects in health economic evaluations. The time frame of the QALY calculations is related to the follow-up time periods.

#### 2.5.8. Physical Activity Scale for the Elderly

The Physical Activity Scale for the Elderly (PASE) is a 10-item questionnaire that measures the frequency of physical activities undertaken in the past seven days. It includes questions on the frequency and duration of various leisure-time, household, and work-related activities [53]. The PASE was developed and validated to assess physical activity in people over 65 [53]. A Swedish version developed by Katharina Stibrant Sunnerhagen for rehabilitation medicine at the University of Gothenburg in 1992 (unpublished material) will be used. The questionnaire consists of 10 questions, where questions 1–6 are about leisure activities, 7–9 are about household activities, and question 10 is about activities related to work. A high score indicates a higher level of physical activity.

#### 2.5.9. Self-Efficacy (Exercise)

Self-efficacy for physical activity is the belief and conviction in one’s ability to be physically active in various circumstances—i.e., an individual’s confidence in overcoming factors that may constitute obstacles to physical activity. One’s belief in one’s ability affects how much effort is put into performing the behaviour, how long one exerts oneself despite any obstacles and setbacks, and how one can start and maintain physical activity [54]. The Self-Efficacy for Physical Activity (SEPA) scale consists of five questions related to barriers to regular physical activity (weather, mood, fatigue, vacation, and busyness). Subjects assess their confidence in participating in physical activity in different situations on a 5-point scale. The scale has been used in several health-promoting studies and evaluated for reliability and validity for diverse populations.

### 2.6. Data Analysis

There will be four points of data analysis: grounded theory, thematic method, descriptive and analytical statistics, and a health economic analysis.

#### 2.6.1. Grounded Theory

Grounded theory is used to analyse individual and focus group qualitative interviews. According to the grounded theory method, data analysis intertwines with the interviews [40]. The study is based on the following steps in grounded theory: First, each transcript will be read several times to develop a sense of the overall context of the data and to gain familiarity with the demographic characteristics and detailed information provided. Second, line-by-line initial open coding on the interview transcripts will be performed manually, and the following questions will be asked: “What is happening”? Additionally, “What are the processes that take place?”. The codes are brought together based on similarities and patterns. During this analysis step, the main concern appears in the data. Third, Analytical reflexive memos will be written to interpret and connect concepts and relationships among codes. Fourth, focused coding will be performed to explore preliminary subcategories with a comparative method. Fifth, theoretical sampling will deepen the analysis and develop and refine the emerging theory’s subcategories and core categories. To validate the theory, the codes, subcategories and core categories will be reviewed and critically discussed repetitively. In this process, various subcategories will be filled in and saturated. When conceptual saturation has been achieved, data collection ends [40].

#### 2.6.2. Thematic Method

Thoughts and notes from the fieldwork constitute a preliminary analytical reflection on the material, further processing, and deepening in the actual analysis process [55]. The field notes of the observations will be analysed using an inductive thematic method [56], where the material is sorted and interpreted for the purpose of the study. The prominent themes will be analysed with the support of a theoretical framework, which depends on the focus of the material [57]. Participating researchers have extensive experience with ethnographic methods.

#### 2.6.3. Descriptive and Analytical Statistics

Descriptive and analytical statistics will be adapted to the data level and distribution. Furthermore, the Bayesian Repeated Measure ANOVA, performed within JASP Team (Version 0.16.4) [58], will evaluate whether there is any change over time regarding health-related quality of life, vision-related quality of life, and physical activity. Researchers with experience in statistical analysis will carry out the analyses. For a comprehensive description of the advantages of applying this analysis in our small sample study over more traditional analysis, see [59].

#### 2.6.4. Health Economic Evaluation

In parallel with the Swedish intervention reported here, a Danish randomised controlled trial will be conducted with the same inclusion criteria for participants but with a more traditional physical activity intervention. The health economic evaluation will explicitly compare two active arms, empowerment-based versus traditional physical activity, and a control group regarding costs and consequences, such as health-related quality of life in the form of QALYs.3. Results

This section may be divided into subheadings. It should provide a concise and precise description of the experimental results, their interpretation, and the experimental conclusions that can be drawn.

## 3. Discussion

The study will be an example of a mixed-method study comprising a multi-case intervention study with qualitative and quantitative analyses but with a qualitative core component. The study protocol, with multiple data sources, is significant for fully capturing the various aspects of what it means to have AMD and generating a comprehensive understanding of the impact of participating in an empowerment-based intervention. EPI includes a participatory strategy and involves the participants in the choices of activities, listening to their needs. The intervention also acknowledges the need for expert opinion and theory in the empowerment and health promotion field to design the intervention. This strategy is essential to Reflective Equilibrium Community Empowerment, which takes advantage of top-down and bottom-up approaches [35].

The bottom-up approach in EPI entails that the participants will be allowed to have as much control as possible over the activities they are involved in and is an essential part of an empowerment process [39]. Since the participants will be coached at the beginning of the intervention, we will be able to adapt the intervention to the participants’ local environment and social context and understand their specific reality. This can contribute to a better condition for effective authorisation [60,61,62]. Empowerment is also seen as a key strategy for sustainable health promotion efforts.

The results of this project will fill a gap in the research literature and may provide insights into the potential of creative supervised, adapted physical activity, health coaching, and socialising. Thus, EPI might increase physical activity and strength and reduce sedentary behaviours for older people with AMD. The social part of the EPI is also aimed at supporting the participants in changing their social lives. This can be particularly important after a pandemic when older people have been isolated for a long time. The EPI is also aimed at strengthening their participation in existing municipal group training for older people, sports movement activities, and the Parasport Association’s activities after completion of EPI, resulting in a sustainable solution for the participants. Thus, EPI could improve the health-related determinants of daily life.

A limitation of the planned study is that the findings will represent a small sample of respondents. A potential risk to the project implementation is the health status of the participants, as they are older people with impaired vision and possible co-morbidities. However, one of the purposes of empowerment-based physical activity is to adapt the activities to individual participant conditions, hopefully decreasing the drop-out rate during the project period. There is also the risk of the loss to follow-up becoming considerable. Thus, the primary endpoint of the trial is at the end of the intervention at six months, as the longer follow-up at 12 months after the end of the intervention poses a risk of a high proportion of missing data.

## 4. Conclusions

The chosen methodology offers a structured way for researchers to explore the experiences and factors that may provide insights into the potential of creative supervised, adapted physical activity in groups, health coaching, and socialising that are significant to enable well-being among older individuals with AMD. If successful, EPI might promote well-being by optimising physical, social, and mental health and developing and maintaining functional ability. Healthy ageing is of great importance and presupposes a multidisciplinary approach since ageing is so multifaceted, which our study considers.

## Figures and Tables

**Figure 1 ijerph-20-00643-f001:**
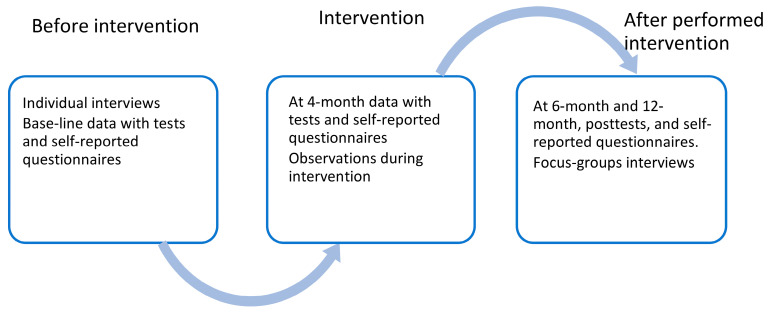
Data collection.

## Data Availability

All data will be coded so that only relevant researchers can access it. Data will be locked in a safe at the university, away from the code key, which will be locked in another safe. Research material is saved for ten years after the research principal completes the project. Only researchers who belong to the project will have access to data. All information that can reveal the identity of the researchers will be pseudonymised.

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
