# Peer review of "Empowerment-Based Physical Activity Intervention for People with Advanced Dry Age-Related Macular Degeneration: Mixed-Methods Protocol"

_ijerph, 2022, doi:10.3390/ijerph20010643_

Round 1

Reviewer 1 Report

Lindgren and colleagues proposed a new protocol to describe a prospective, 12 mixed-methodology for studying a population with AMD before, during, and after an empower-13 ment-based physical activity intervention (EPI). The protocol paper is well written and the hypothesis are explained very well. 

Congratulations for your protocol paper!

I have only a minor comment

to give the reader a broader view, I would cite in the introduction that there are psychophysical trainings (perceptual learning) with the goal to improve peripheral visual functions in these people. 

These articles may be interesting also for the authors (they don't have to cite all of them, but i think they can be interesting for the authors) 

Maniglia, M., Pavan, A., Sato, G., Contemori, G., Montemurro, S., Battaglini, L., & Casco, C. (2016). Perceptual learning leads to long lasting visual improvement in patients with central vision loss. Restorative neurology and neuroscience34(5), 697-720.

Maniglia, M., Cottereau, B. R., Soler, V., & Trotter, Y. (2016). Rehabilitation approaches in macular degeneration patients. Frontiers in Systems Neuroscience10, 107.

Maniglia, M., Soler, V., Cottereau, B., & Trotter, Y. (2018). Spontaneous and training-induced cortical plasticity in MD patients: Hints from lateral masking. Scientific Reports8(1), 1-11.

Maniglia, M., Cottereau, B., Soler, V., & Trotter, Y. (2016). Facilitatory lateral interactions in patients with age-related macular degeneration. Journal of Vision16(12), 954-954.

Maniglia, M., & Seitz, A. R. (2018). Towards a whole brain model of Perceptual Learning. Current opinion in behavioral sciences20, 47-55.

Maniglia, M., Soler, V., & Trotter, Y. (2020). Combining fixation and lateral masking training enhances perceptual learning effects in patients with macular degeneration. Journal of Vision20(10), 19-19.

Author Response

Dear Reviewer,

Thank you so much for the constructive feedback on our manuscript:

Empowerment-Based Physical Activity Intervention for People With Advanced Dry Age-Related Macular Degeneration: Mixed-Methods Protocol

The changes are made as requested. The answers to the feedback are found in the table.

Kind regards

The authors

Comments

Authors’ answers

Give the reader a broader view, I would cite in the introduction that there are psychophysical trainings (perceptual learning) with the goal to improve peripheral visual functions in these people. 

Thank you so much for presenting the articles by Maniglia et al for us! We have some of them in the introduction in order to give the readers a broader view.

Reviewer 2 Report

The authors present an interesting approach of an empowerment-based physical activity intervention for people with age-related macular degeneration, planning to enrol 20 patients for a six-months duration. The approach addresses the large variability of the elderly population with regard to physical and mental fitness with a mostly qualitative approach. However, a generalizable conclusion can not be expected with such a diverse sample and open approach. Therefore, the authors also implement certain validated questionnaires and physical activity measurments. 

From a clinicians perspective I would wish for a more precise research question. I therefore suggest to focus on the validated measurements as primary outcome variables. Therefore, a power-analysis should be performed with the mentioned pilot study.

Furthermore: 

1. Causalities and correlations are confused

1.1.1: the vague qualitative approach dominates. Measurable parameters are only introduced in 2.4. Both aspects deserve early mentioning and reasoning in this section

1.1.2: Similar criticism as above

2.1: the experience from the established reference group has influence on the expected effect size and power of the study and needs mentioning or reference. 

2.3: the tenets of the declaration of helsinki need to be followed

Section 2.4 clarifies some points mentioned above. 

2.5.1 and 2.5.2: Grounded theory is not a known concept in medical literature and could be explained. is data collection and analysis performed manually or software based? When is the theoretical framework built?

3: I suggest choosing a quantitative core component over the qualitative, but support the empowerment based approach.

General: Title (AMD) does not match study pupulation (advanced dry AMD)

Author Response

Dear Reviewer,

Thank you so much for the constructive feedback on our manuscript:

Empowerment-Based Physical Activity Intervention for People With Advanced Dry Age-Related Macular Degeneration: Mixed-Methods Protocol

The changes are made as requested. The answers to the feedback are found in the table (which you find in the file).

Kind regards

The authors

Round 2

Reviewer 2 Report

Dear Authors and Editor,

Thank you for your comments and the clarifications applied in the manuscript. I still appreciate the attempt to bridge the gap between social sciences and its approaches towards a very important topic in both ophthalmology and also public health. I'm looking forward to see the results of the study.

However, in my personal opinion, the focus on the qualitative approach despite collecting validated quantitative data, does not allow publication in primarily medical journal. I suggest reframing the research question towards the primarily quantitative data. If this is not wished to do, I propose submitting the paper in a social science journal.